# Inference-time Correction of Errors in AI-Generated Chest X-ray Radiology Reports

## Abstract

Automated radiology report generators are being increasingly explored in clinical workflow pilots, particularly for chest X-ray imaging. However, their factual correctness with respect to the description of the findings has often been less than accurate, making their adoption slow and requiring detailed verification by clinical experts. In this paper, we propose an automatic report correction method that uses both image and textual information in automated radiology reports to spot identity and location errors in findings through fact-checking models. Prompts for a pre-trained large language model are then generated from the analysis of these errors to produce corrected sentences by selectively modifying target findings described in the automated report sentences. We show that this method of report correction, on the average, improves the report quality between 17-30% across various SOTA report generators over multi-institutional chest X-ray datasets.

## 1 Introduction

Radiology practices are piloting automated radiology report generator tools for expediting and streamlining structured report generationSyeda-Mahmood et al. (2020). Such reporting tools have progressed the most in chest X-ray radiology thanks to the availability of relatively large datasets such as MIMICJohnson et al. (2019a) and CheXpertIrvin et al. (2019) that come with their companion reports for training vision-language generative (VLM) modelsBannur et al. (2024); Guo et al. (2018); Krause et al. (2017). However, the results with pilots are revealing a predominance of hallucinations and factual errors which have hampered their adoption in clinical workflows. While these tools continue to be improved, there will still be a need for a fact-checking and correction model that can work with deployed and frozen report generators at inference time as a last checkpoint before the information being presented to clinicians.

In this paper, we present a report correction method with a built-in discriminative image-guided fact-checking (FC) model that detects and localizes the errors in the report. The error analysis along with the report sentences is used to generate a corrective prompt to an LLM which then produces the corrected sentence. We show that this method of report correction improves the report quality of report generators between between 17-30% across various SOTA report generators over multi-institutional chest X-ray datasets.

Figure 1d illustrates report correction by our method for an automatically generated report in Figure 1 using both the chest X-ray image (Figure 1a) and structured finding descriptions derived from the automated report in Figure 1c. The result is an improved match to the ground truth report of Figure 1b.

Our approach is based on 3 key insights. First, a fact-checking examiner model that has the authority to find and correct errors in automated AI reports must be developed independent of the techniques used to develop reporting models, meaning it cannot be based on LLMs. Secondly, it should still cover the space of possible error instances made by such report generators, even if restricted to known types of errors without requiring data on instances of errors made by automatic report generators to be available. Such data wold be difficult to acquire needing not only access to all report generators but also large variety of clinician-annotated ground truth datasets to catalog the errors. Next, the correction must be done in a conservative way weighing the self-consistency of the examiner to account for the eventuality that the examiner model itself makes a mistake. Finally, the report correction should lead to an overall improvement in the quality of the report.

| Chest X-ray | Ground Truth Report | Automated Report | Corrected Report |
|---|---|---|---|

Figure 1: Illustration of report correction. (a) Chest X-ray image. (b) A section of its ground truth radiology report. (c) Automatically generated report by XrayGPTThawkar et al. (2023). (c) Corrected report by our method. The sentence with error in finding is colored orange in (c) and corrected sentence is shown in green in (d). Here the erroneous finding of "pleural effusion" is removed while still retaining location information for the remaining finding in the sentence, i.e. atelectasis.

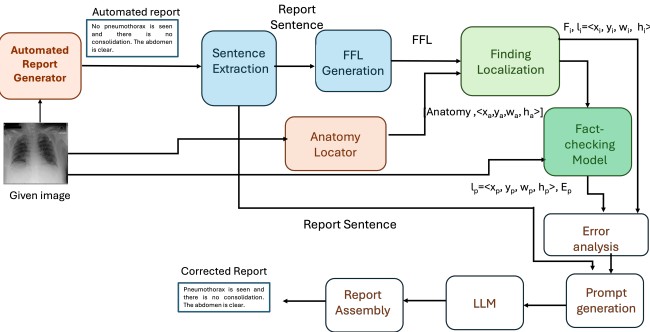

Figure 2: Illustration of the report correction workflow using a fact-checking model-guided LLM.

## 2 RELATED WORK

While there is considerable work in chest x-ray radiology report generation literatureBannur et al. (2024); Endo et al. (2021); Gao et al. (2024); Li et al. (2019); Pang et al. (2023); Ramesh et al. (2022); Ranjit et al. (2023); Syeda-Mahmood et al. (2020), papers focusing on detecting and correcting errors in radiology report generation have only recently been emerging for inference-time fact-checkingMahmood et al. (2023). However, the correction approach has been to simply remove the entire sentence. Standard approaches of hallucination reduction through direct policy optimization (DPO)Hardy et al. (2024); Passi & Shah (2022); Rafailov et al. (2023); Zhou et al. (2023) or proximal policy optimization (PPO)Zheng et al. (2023); Ziegler et al. (2019) are not applicable at clinical inference time. Other inference-time fact-checking methods that consult external knowledge sources cannot be used for patient-specific radiology reports either. Lab; Passi & Shah (2022); Suprem & Pu (2022). Even powerful LLM-as-a-judge models are not often trained for such domain and patient-specific applications, to be reliable enough in the role of the examiner. Thus, to our knowledge, combining fact-checking models with large language models for radiology report correction, has not been previously attempted.

## 3 REPORT CORRECTION METHOD

The overall report correction process is illustrated in Figure 2. A report produced by an automated report generator for chest X-rays is pre-processed to extract sentences, and findings from sentences. The extracted findings are structured as fine-grained label (FFL) patternsSyeda-Mahmood et al. (2020), documenting the presence or absence of a finding and any associated anatomical location information. A finding localization algorithm is then used to extract an indicated anatomical image location $l_i = <x_i, y_i, w_i, h_i>$ for the finding from the report. A fact-checking model uses the image $I$, and the finding pattern $F_i$ to predict an expected location $l_p = <x_p, y_p, w_p, h_p>$ and a veracity label $E_p$ for $F_i$. The spatial overlap error between the predicted and indicated location along with the veracity indicator $E_p$ is used to generate distinct prompts for different actions in the error analysis module. These are submitted to a large language model (LLM) to perform the sentence

| No | Sentence | FFL |
|----|----------|-----|
| 1. | FINDINGS: The heart appears mildly enlarged. | anatomical finding\|yes\|enlarged cardiac silhouette\|heart |
| 2. | Cardiac size is slightly enlarged allowing for limitations of this AP view. | anatomical finding\|yes\|enlarged cardiac silhouette\|heart |
| 3. | Pleural vasculature is not engorged and the patient has moderate pulmonary edema on the right. | anatomicalfinding\|no\|vascular congestion\|lung |
| | | anatomicalfinding\|yes\|pumonary edema\|lung\|right |

Table 1: Illustration of structured finding extraction using the FFL pattern extraction algorithmSyeda-Mahmood et al. (2020).

correction. The corrected sentences along with valid sentences from the report are combined in order to assemble the overall corrected report.

### 3.1 SYNTHESIZING THE SPACE OF FINDING ERRORS

We focus on modeling the most common types of errors made in radiology reports, which are false predictions, omissions, and incorrect finding location reportingRao et al. (2025); Yu et al. (2023). To ensure coverage of instances of these errors in AI reports during synthesis, the set of findings seen in chest X-rays must be known and captured in a structured way to enable synthesis. Fortunately, previous work has already cataloged all clinically significant findings in chest X-rays in a chest X-ray lexiconWu et al. (2020). Further, algorithms are available that reliably extract the findings from report sentences in the form of structured patterns called fine-grained finding patterns (FFL) which normalize them to the standard vocabulary from the chest X-ray lexiconSyeda-Mahmood et al. (2020). We chose the FFL extraction algorithm as it could detect the largest number of findings (78 core findings and 101,088 distinct FFL patterns Wu et al. (2020)) with over 97% accuracySyeda-Mahmood et al. (2020). Using this algorithm, a finding $F_i$ is described in a structured way as:

$$F_i = T_i|N_i|C_i|A_i|L_i \tag{1}$$

where $T_i$ is the finding type, $N_i =$ yes\|no indicates a present or absent finding respectively, $C_i$ is the normalized core finding name, $A_i$ is the anatomical location, $L_i$ reflects laterality of the core finding $C_i$. In this paper, we use $F_i$ to refer to the full FFL pattern as in Equation 1 as well its shortened form $N_i|C_i$ as appropriate. The FFL pattern is a normalized way to describe the finding using standard vocabulary as shown for sentence 1 and 2 in Table 1 for cardiomegaly.

To synthesize the finding locations, we use an anatomical localization algorithm that locates all distinct anatomical regions known to contain chest X-ray findings through bounding boxes Wu et al. (2021a). This algorithm detects the largest number of anatomical regions (36 regions) with average localization precision and recall of 0.896 and 0.881 respectivelyWu et al. (2021a) and was used to generated the ChestImaGenome dataset for MIMIC imagesJohnson et al. (2019a). The findings are then localized by merging the bounding boxes of the relevant anatomical regions covered by the finding as given by the clinical knowledge in the chest X-ray lexiconWu et al. (2021a). Although this method can over or underestimate the precise boundary of a finding, since locations are only roughly described in radiology reports, this is sufficient for report verification. We rely on clinician-corrected bounding box locations, however, during training the fact-checking model to enable higher precision in localization.

We assembled a large ground truth dataset of chest X-ray images with their associated clinician-produced radiology reports reflecting over 78 clinically significant findings. Structured finding descriptors (FFL) and anatomical locations of findings were extracted. We then derived a synthetic dataset of correct and incorrect pairings of images with findings by mixing and matching findings of one image with the another allowing us to create a very large synthetic dataset of over 24 million pairs. Since the findings were derived from clinical knowledge rather than their occurrence in automated reports, all major error combinations made by report generators are guaranteed to resolve to these findings, thus ensuring sufficient coverage of the finding combinations seen in automated reports.

Specifically, let $< I, R >$ be a sample set of ground truth image-report pairs in a publicly available dataset $D$. Let $F = \{F_j\}$ be the total list of possible findings across chest X-ray datasets. Given a real finding $f_{ij}$ at location $l_{ij}$ for a sample image-report pair $D_i$, we create 3 variants to reflect (a) reversal of polarity (b) relocation of the finding (c) and substitution with appropriate relocation as $FL_{iincorrect} = \{< \overline{fl_{ij}}, fl_{ik}, fl_{mn} >\}$, where $\overline{fl_{ij}}$ is the reversed finding, $fl_{ik}$ is finding $f_{ij}$ relocated to a new position $l_k \in L_j$, and $fl_{mj}$ is obtained by substituting finding $f_j$ with $f_m \notin F_i$ at location $l_n \in L_m$.

Randomly selecting findings and choosing to vary their locations can create a large variety of combinations. However, to cover both physically plausible (correct/real) as well as impossible combinations (incorrect/fake), we mine the finding statistics in ground truth reports to derive conditional probabilities of co-occurrence of findings. We then adopt a Monte Carlo sampling strategy to introduce randomness in the synthesis process so that those findings that are likely to co-occur frequently do not bias the generation. As a result of this sampling, each data item can be described by the tuple $< I, F, < x, y, w, h, E >>$ where $I$ is the image, $F$ is an FFL pattern, $< x, y, w, h >$ is the bounding box assigned to the finding $F$ and $E$ is a binary label indicating correct/incorrect nature of the findings with $E = 1$ denoting a correct finding.

## 3.2 DESIGNING THE FACT-CHECKING MODEL

Our fact-checking model is a multi-modal, multi-label supervised contrastive regression network consisting of a feature learner and a regressor as shown in Figure 3. The feature learner is a contrastive encoder that learns a joint representation of images and short FFL patterns. The regressor learns the association of the combined image-text features with the locations of the findings in the image. Throughout, a supervision label of correct or incorrect association $E$ guides the learning.

**Feature learning**

A natural choice for a multimodal contrastive encoder is a vision language model such as CLIPRadford et al. (2021). However, unlike CLIP, instead of a single positive image-text pair, we have multiple such pairs corresponding to the findings reported as present or absent in the image. Further, all other pairings are not considered negative as in CLIP since some findings may not even be reported (i.e. are unknown or not important enough to report). Unlike the self-supervision provided by the pairs in CLIP, we have additional supervision coming from the $E$ label indicating the correctness of the finding and location. This results in a non-diagonal similarity matrix for our feature encoder as shown in Figure 3. To train this similarity matrix, we define a multi-label cross-modal supervised contrastive loss function as:

$$\mathcal{L}_{SupC_i} = \frac{-1}{|F_{icorrect}|} \sum_{f_{ij} \in F_{icorrect}} log \frac{e^{s_{if_{ij}}/\tau}}{\sum_{a_{ik} \in F_{iincorrect}} e^{s_{ia_{ik}}/\tau}} \tag{2}$$

where $s_{if_{ij}} = z_i \cdot z_{f_{ij}}$ is the pairwise cosine similarity between image and textual embedding vectors from the correct findings $f_{ij} \in F_{icorrect}$, and $s_{ia_{ik}} = z_i \cdot z_{a_{ik}}$ is with the incorrect findings where $a_{ik} \in F_{iincorrect}$. The overall loss is obtained by averaging across all the samples in the batch. Here $\tau$ is the temperature parameter. Note that unlike the usual supervised contrastive loss functionKhosla et al. (2020), the summation in the denominator is only over the incorrect findings instead of all negative pairs, thus resulting in a new loss function.

**Regression network**

The joint embedding space of the feature encoder is not directly suitable for separating the correct from incorrect finding-image associations as the cosine similarity values between their encodings overlap completely. Instead, we found that by forming a high-dimensional feature space by concatenating the contrastively learned image and text embeddings results in better separability between correct and incorrect pairings. The regression classifier, therefore, is a neural network that takes the projected joint embeddings $T_{ijcorrect} = [z_i | z_{f_{ij}}]$ of image $I_i$ paired with correct finding label $f_{ij} \in F_{icorrect}$ or incorrect labels $T_{ijincorrect} = [zi | z_{a_{ik}}]$ where $a_{ik} \in F_{iincorrect}$ and the corresponding supervision label $Y_g = < Y_{1g}, Y_{2g} >$ where $Y_{1g} = < x, y, w, h >$ is the location and $Y_{2g} = E = 1$ for the real finding and 0 otherwise. Using $Y_p = < Y_{1p}, Y_{2p} >$ as the prediction from the network, we can express the regression loss per sample as a combination of an MSE loss

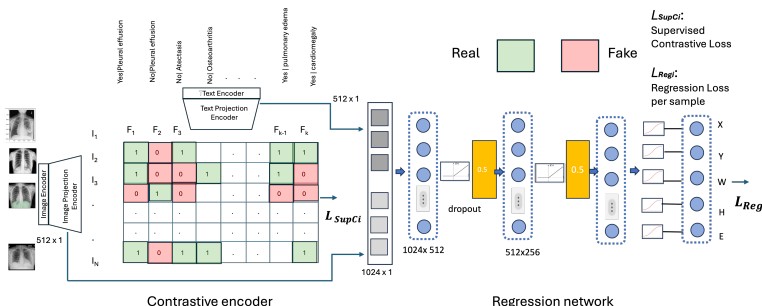

Figure 3: Illustrative of the multimodal supervised contrastive regression network. Here the feature extractor is a supervised contrastive encoder training with a non-diagonal similarity matrix. The classification network is a regressor on both the location and veracity of the label using the combined image and textual input from the finding pattern.

measuring the spatial overlap in location and a binary cross-entropy loss for the correctness of the predicted finding, reflecting the dual attributes being optimized as:

$$\mathcal{L}_{Reg_i} = \underbrace{|Y_{1p} - Y_{1g}|^2}_{\mathcal{L}_{Spatial_i}} - \underbrace{[Y_{2g}\mathbf{log}(Y_{2p}) + (1 - Y_{2g})\mathbf{log}(1 - Y_{2p})]}_{\mathcal{L}_{Identity_i}} \tag{3}$$

**End-to-end training the FC Model**

Bringing these two networks together, the fact-checking model was trained as a single end-to-end learning network as shown in Figure 3. The encoder model was based on a chest X-ray pre-trained CLIP and reused its image and text encodersRamesh et al. (2022). The joint embedding projection layers of this model (768x512 for image and 512x512 for text) were, however, fresh-trained using the new supervised contrastive loss mentioned in Equation 2. The regression network (657,413 parameters) consists of two linear layers, two drop out layers followed by a RELU for the intermediate layers and separate sigmoidal functions for producing the output regression vectors as shown in Figure 3. The losses defined in Equations 2 and 3 were applied at the respective heads with the backpropagation for the regression loss going back into the contrastive learning part as well. The total trainable parameters were 151,277,313 parameters making it possible to build this model on an NVIDIA A100 GPU with 40GB of memory. The network was trained for 100 epochs using the AdamW optimizer with a batch size of 32. The cosine annealing learning rate scheduler was used with the maximum learning rate of 1e-5 and 50 steps for warm up.

### 3.3 REPORT CORRECTION

To correct the reports, the output of the FC model is analyzed. Given an indicated finding $F_i$ extracted from the automated report associated with a given image $I$ at inference time, it can predict a location $l_p = <x_p, y_p, w_p, h_p, E_p>$. Using the finding localization algorithm of Section 3.1, we can also derive the finding's indicated location as $l_i = <x_i, y_i, w_i, h_i>$. The corrective action rules are formed both using the predicted veracity indicator $E_p$ and the spatial overlap between $l_i$ and $l_p$ measured through IOU as

$$\overline{IOU}_{pi} = 1 - IOU_{pi} = 1 - \frac{|l_p \cap l_i|}{|l_p \cup l_i|} \tag{4}$$

Given the possible values of $l_p, E_p, F_i, \overline{IOU}_{pi}$, there could be a large number of error cases to consider. To simplify the analysis, we quantized these values into ranges. For $F_i$ we consider two major classes of findings, namely, presence findings and absence findings as the location indicators are very different for these. The absence findings are associated with the location coordinates $< 0,0,0,0 >$ in both $l_i$ and $l_p$ if predicted correctly. Thus the values of $l_p$ could be categorized into two categories if $l_p \approx 0 = < 0,0,0,0 >$ or $> 0$. The veracity label $E_p$ is already a binary indicator. Similarly, $\overline{IOU}_{pi}$ can be thresholded by a parameter $\Gamma$ to indicate a small difference in the spatial location ($\overline{IOU}_{pi} \leq \Gamma$) or not. Here we choose $\Gamma = 0.01$ in normalized image coordinates as that

Table 2: Illustration of error analysis using the output of the FC model. The error interpretation and suggested corrective action for a finding $F_i$ mentioned in the sentence $S_i$ are shown in the table.

| $l_p$ | $E_p$ | $F_p$ | $\overline{IOU_{pi}}$ | Interpretation | Corrective Action | Prompt |
|---|---|---|---|---|---|---|
| $\approx 0$ | 1 | Absence | $<= \Gamma$ | Both finding and location are correct. | Do nothing as it is correct. | None |
| $> 0$ | 0 | Absence | $> \Gamma$ | Finding is present as per FC. | Flip the finding from absence to presence. Leave the location unspecified. | Remove "no $< F_i >$" and add "yes $< F_i >$" in the sentence: $< S_i >$ |
| $\approx 0$ | 0 | Presence | $<= \Gamma$ | FC Model is saying finding is absent | Flip the finding from present to absent. Leave the location unspecified as it is either close or unspecified already. | Remove "yes $< F_i >$" and add "no $< F_i >$" in the sentence: $< S_i >$ |
| $\approx 0$ | 0 | Presence | $> \Gamma$ | FC model is saying finding is absent | Flip the finding from present to absent. Remove location hint since the location is far away. | Remove "yes $< F_i >$", add "no $< F_i >$", and remove location $< A_i >$ from the sentence: $< S_i >$ |
| $> 0$ | 1 | Presence | $<= \Gamma$ | Both finding and location are correct. Finding is a presence finding | Do nothing as it is correct. | None |
| $> 0$ | 1 | Presence | $> \Gamma$ | Finding is correct and present but location is wrong | Drop location only. Keep the finding. | Remove location $< A_i >$ from the sentence: $< S_i >$ |
| All other combinations. | | | | Either $E_p$ or $l_p$ is incorrect. | Do Nothing as FC Model itself is incorrect. | None |

was empirically found to be the gap between anatomical regions in chest X-ray regional annotations. With this quantization, we have 2 x 2 x 2 x 2 = 16 possible combinations to analyze for errors. Of these, the combination $(L_p = 0, E_p = 1, \overline{IOU_{pi}} > \Gamma)$ is impossible for an absent finding since its location is not mentioned in reports. Of the 15 combinations, 6 correspond to consistent output from the FC model. These were manually analyzed to arrive at an interpretation and a corrective action, from which 5 unique prompt templates were designed as shown in Column 6 of Table 2. The remaining combinations were potential inconsistency cases in the prediction of the FC model itself. While the FC model performed well across the datasets tested, a potential error in the FC model could potentially worsen the report quality. Fortunately, because we regressed on both location and veracity, we can spot such inconsistencies through these combinations to conservatively disable any corrective action. For example, a combination of $(L_p = 0, E_p = 0, \overline{IOU_{pi}} \leq \Gamma)$ for an absent finding $F_i$ is a case where either the location prediction or the veracity indicator is incorrect.

**LLM-based sentence correction**

Given the FFL patterns and sentences extracted from automated reports, instances of prompts are obtained using the prompt templates indicated in Column 6 Table 2 and given to a large language model to initiate sentence modification and correction. Any well-trained LLM would be sufficient for our purpose as these days, they can all be instruction-tuned for sentence correction, and their choice mainly effects the readability of the report rather than the finding and its description. Nevertheless, we used Llama3.2 as it was freely available and fit within the GPU size of our server. Since the average sentence in a report has 13-15 words, and sentence correction task is fairly deterministic, we used 400 token limit with a temperature of 0 for sentence correction. The sentence returned by the LLM are then assembled to form the corrected report. Since duplicate sentences could arise from multiple findings being edited in a given sentence, they are detected and removed. Depending on the order of findings edited, the actual sentences in the corrected report may come in a different order than the automated report, which can also be corrected in a post-processing final step of assembly.

## 4 RESULTS

We now report our evaluation of the report correction approach using multiple benchmark datasets and report generators.

**Datasets used and created**

We selected several publicly available multi-institutional datasets of chest X-ray images annotated for findings and their locations as summarized in Table 3. All datasets were clinician validated and

Table 3: Details of the datasets used in experiments.

| Dataset | Patients Train/Val/Test | Images | Findings | Regions | Real/Synth. |
|---|---|---|---|---|---|
| RadCheckMahmood (2025) | 44,133/6,274/12,538 | 243,311 | 49 | 922,295 | 1.616M/27.047M |
| CImaGenomeGWu et al. (2021a) | 288/33/69 | 461 | 35 | 5,477 | 4,063/23,463 |
| MS-CXRJohnson et al. (2019a) | 478/54/114 | 925 | 8 | 2,254 | 2,247/24,338 |
| ChestXray8Wang et al. (2017) | 457/51/109 | 880 | 8 | 1,571 | 1,571/10,137 |
| VinDr-CXRNguyen et al. (2022) | 9,450/1,050/2,250 | 15,000 | 23 | 69,052 | 47,973/132,632 |

Table 4: Illustration of LLM-based report correction showing the original sentence, the incorrect finding, the actual LLM prompt generated, and the resulting corrected sentence.

| Original Sentence | Incorrect finding | LLM Prompt (w/o sentence) | LLM-Corrected Sentence |
|---|---|---|---|
| Left-sided pleural effusion found and the right atelectasis still remains. | yes\|pleural effusion | Remove "pleural effusion" from sentence: | Right atelectasis still remains. |
| The chest x ray image shows no focal consolidation, pulmonary edema, pleural effusion or pneumothorax | no\|pneumothorax | Remove "no pneumothorax" from the sentence: | The chest X-ray image shows no focal consolidation, pulmonary edema, pleural effusion. |
| The chest x ray image shows no focal consolidation, pulmonary edema, pleural effusion or pneumothorax | no\|pneumothorax | Remove "no pneumothorax" and add "yes pneumothorax" in the sentence: | The chest X-ray image shows no focal consolidation, pulmonary edema, or pleural effusion, but shows pneumothorax. |
| There is left pleural effusion and pneumothorax | yes\|pleural effusion | Remove "left pleural effusion" but keep the "left pneumothorax" in the sentence: | There is left pneumothorax. |
| The chest x ray image shows no left focal consolidation. | no\|consolidation | Remove "left" from the sentence: | The chest x ray image shows no focal consolidation. |

Table 5: Illustration of FC model evaluation under multiple datasets. The last two rows show comparison with two SOTA methods for fact-checking and phrasal grounding. Here CG stands for ChestImaGenome Gold, C8 is ChestXray-8, MCXR is MS-CXR, and VinDr stands for VindDr-CXR.

| Method | Metric | CG | MCXR | C8 | VinDr |
|---|---|---|---|---|---|
| FC Model | Accuracy | 0.92 | 0.94 | 0.92 | 0.90 |
| FC Model | MIOU | 0.54 | 0.53 | 0.57 | 0.49 |
| R/F Model | Accuracy | 0.84 | 0.78 | 0.81 | 0.83 |
| Maira-2 | MIOU | 0.39 | 0.48 | 0.51 | 0.42 |

Table 6: Illustration of the report quality improvement using fact-checking guided LLM using various report quality metrics. Here RadF1 stands for Radgraph F1.

| Generator | RadF1 | | RQ | | BLEU | | SBERT | |
|---|---|---|---|---|---|---|---|---|
| | (A,G) | (C,G) | (A,G) | (C,G) | (A,G) | (C,G) | (A,G) | (C,G) |
| RGRGTanida et al. (2023) | 0.52 | 0.67 | 0.46 | 0.52 | 0.24 | 0.29 | 0.33 | 0.43 |
| XrayGPTThawkar et al. (2023) | 0.39 | 0.45 | 0.37 | 0.48 | 0.14 | 0.24 | 0.26 | 0.38 |
| GPT4-in | 0.43 | 0.51 | 0.35 | 0.47 | 0.11 | 0.19 | 0.09 | 0.14 |
| R2GenGPTWang et al. (2023) | 0.54 | 0.58 | 0.37 | 0.49 | 0.19 | 0.27 | 0.38 | 0.47 |
| CV2GPT2Nicolson et al. (2023) | 0.41 | 0.49 | 0.38 | 0.48 | 0.14 | 0.24 | 0.43 | 0.54 |
| CheXRepairRamesh et al. (2022) | 0.38 | 0.43 | 0.36 | 0.44 | 0.21 | 0.28 | 0.39 | 0.46 |
| Maira-2Bannur et al. (2024) | 0.58 | 0.63 | 0.52 | 0.59 | 0.20 | 0.26 | 0.43 | 0.51 |
| Avg.Improv. | 13.5% | | 27% | | 48.2% | | 32.5% | |

vetted for bias and fairness during their IRB approval. For training the fact-checking model, we created a synthetic dataset as described in Section 3.1 starting from the ChestImaGenome Silver datasetWu et al. (2021b) which in turn was derived from MIMIC-CXRJohnson et al. (2019b). The resulting dataset called RadCheck contains over 24 million samples of image pairings with both correct and incorrect finding-location descriptions and is now available in open source on HuggingfaceMahmood (2025). Finally, as other datasets listed in Table 3 already provided findings and locations without ground truth reports, we used the same mixing and matching methodology specified in Section 3.1 to create the correct and incorrect pairings for our evaluations experiments. The testing partitions of the datasets were used for the evaluations, while the training partition of RadCheck was used for training the FC model.

**Report generators**

We also selected 7 SOTA automated report generators whose Github code was freely available. These included MAIRA-2Bannur et al. (2024), ChexRepairRamesh et al. (2022), RGRGTanida et al. (2023), XrayGPTThawkar et al. (2023), R2GenGPTWang et al. (2023), CV2DistillGPT2Nicolson et al. (2023) and our in-house hospital implementation of GPT-4 (GPT4-inhouse). These included automated report generation methods that are based on the latest LLava-style VLM models, with varying capabilities including phrasal ground (RGRG), multi-view and longitudinal information handling (MAIRA-2), and distillation-based models.

**Finding error detection performance**

We evaluated the accuracy of FC model in finding veracity prediction and localization using the test partitions of the datasets shown in Table 3. The performance was seen to remain stable for different datasets with the model consistently yielding an accuracy over 90% for correct/incorrect finding classification, as shown in Table 5. By using 10 fold cross-validation in the generation of the (70-10-20) splits for the datasets, the average accuracy of the test sets lay in the range $0.92 \pm 0.12$. In addition, we measured the spatial localization performance through mean IOU measure of spatial overlap between the predicted and ground truth bounding boxes of finding provided in the datasets. This was found to lie in the range 0.49-0.57, indicating that the predicted locations of findings from the fact-checking model have at least 50% overlap with the ground truth finding locations.

**Comparison to other methods**

With no prior work on fact-checking with phrasal grounding for chest X-ray reports, we compared to the nearest methods that either do phrasal grounding Maira-2Bannur et al. (2024)) or real/fake classification (the R/F Model from Mahmood et al. (2023)). The results are shown in Table 5 with the last two rows recording the relevant numbers for a regressor or classifier respectively showing that the FC Model outperforms both these methods across all the datasets.

**Report correction performance**

Using an LLM to correct report sentences based on the corrective action templates provided in Table 2 resulted in well-formed sentences with the erroneous portions removed. Table 4 shows examples of report sentences corrected through the LLM in this manner. As can be seen, the resulting sentences are properly formatted language-wise, and reflect the intended corrective action.

To objectively measure the performance improvement across report generators, we ran the report generation tools on the test partitions of all the datasets. We then extracted the findings (FFL patterns) and their anatomical locations as described in Section 3.1. A similar processing was applied to the corrected reports and the ground truth reports when available.

**Report quality improvement across metrics**

We then recorded the report quality improvement by noting the difference in similarity between automated report (A) to the ground truth report (A,G), versus the similarity between corrected report (C) and the ground truth report (C,G). The similarity between two reports was measured using several metrics, selecting representative methods from lexical word overlap scores (BLEUPapineni et al. (2002)), semantic textual matching (SBERTZhang et al. (2019)), clinical accuracy F1-score Jain et al. (2021), and phrasal-grounded accuracy such as RQMahmood et al. (2025). We used the Chest ImaGenome Gold dataset for this experiment as it had ground truth report with clinician validated findings. The resulting values of these metrics across the report generators for this dataset are shown in Table 6. This table indicates that the report quality improved across all report generators

Table 7: Illustration of report quality improvement using RQ score across various datasets and report generators. In each case, the corrected report (C) shows higher similarity to the ground truth report (G) than the automated report. Here CG=ChestImaGenome Gold, C8=Chest-Xray8, and VinDr=VinDr-CXR datasets.

| Generator | CG | | MCXR | | C8 | | VinDr | |
|---|---|---|---|---|---|---|---|---|
| | RQ | | RQ | | RQ | | RQ | |
| | (A,G) | (C,G) | (A,G) | (C,G) | (A,G) | (C,G) | (A,G) | (C,G) |
| RGRGTanida et al. (2023) | 0.46 | 0.52 | 0.51 | 0.62 | 0.38 | 0.49 | 0.51 | 0.63 |
| XrayGPTThawkar et al. (2023) | 0.37 | 0.48 | 0.45 | 0.49 | 0.35 | 0.42 | 0.46 | 0.54 |
| GPT4-inhouse | 0.35 | 0.47 | 0.46 | 0.54 | 0.41 | 0.48 | 0.51 | 0.58 |
| R2GenGPTWang et al. (2023) | 0.37 | 0.49 | 0.44 | 0.54 | 0.38 | 0.47 | 0.51 | 0.57 |
| CV2DistillGPT2Nicolson et al. (2023) | 0.38 | 0.48 | 0.39 | 0.49 | 0.41 | 0.47 | 0.52 | 0.6 |
| CheXRepairRamesh et al. (2022) | 0.36 | 0.44 | 0.45 | 0.51 | 0.43 | 0.49 | 0.51 | 0.59 |
| Maira-2Bannur et al. (2024) | 0.52 | 0.59 | 0.47 | 0.58 | 0.41 | 0.49 | 0.50 | 0.61 |
| Avg. Impv. | 13.5% | | 18.7% | | 19.14% | | 16.5% | |

independent of which metric was used for comparison with improvements ranging from 13.5%-48.2% across the metrics and an average around 30.5% improvement seen for this dataset.

**Report quality improvement across datasets**

Finally, we evaluated the generalization of the report quality improvement performance across multiple datasets and report generators. Since some of the metrics (BLEU, SBERT) needed full ground truth reports which were not available for all datasets, we focused the evaluation using the RQ score as it utilized the finding as well as location information in the provided ground truth across datasets. The resulting performance of the 7 report generators tested across 4 datasets is shown in Table 7. Since RQ score recorded agreement in the finding identity and spatial overlap in the locations of findings, it was able to capture the combined improvement in report quality well across all datasets for all report generators tested, averaging an improvement around 17% across the datasets as shown in that table.

**Limitations**

Although our work is the first to date to correct radiology reports in this automated way, it does have limitations. Due to limited scope, it does not address severity and measurement errors relating to findings. Secondly, the corrections can be applied to only mentioned findings in reports while missed mentions cannot be added to the report. Next, potential errors in finding extraction and localization could lead to prediction error in the FC model and inconsistencies in error interpretation leading to the selection of incorrect prompts. Finally, the phrasal grounding is currently using bounding boxes which only approximately localize a finding. Full-fledged segmentation of findings may lead to better results. Due to space limitations, we have not reported here the performance of our model in terms of the type of finding errors and their criticality. Finally, the LLM-based report correction can be continually improved with the design of more specific prompts per finding further specializing the templates. Since their output is not guaranteed to be the same in each run, variability could still exist in the reports. These issues will be addressed in future work.

## 5 CONCLUSIONS

In this paper, we have presented a novel method of correction of generative AI reports for chest X-rays by focusing on findings. We developed a fact-checking model covering a large fraction of finding errors and interpreted its output to carve out a set of corrective actions and suitable prompts to result in a higher quality report. Working across data sets and report generators, we have shown an average improvement in report quality ranging from 17-30% across report generators. We hope that such a report correction approach can expedite the adoption of AI reporting models in clinical workflows in future.

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
