# OpenReview forum: "Inference-time Correction of Errors in AI-Generated Chest X-ray Radiology Reports"
_ICLR.cc/2026/Conference — ICLR 2026 Conference Withdrawn Submission_

### Official Review · Reviewer_oJgk · 2025-10-30

**Soundness:** 3
**Presentation:** 1
**Contribution:** 2
**Rating:** 4
**Confidence:** 4

**Summary:**

This paper introduces a novel inference-time report correction framework that improves the factual accuracy of AI-generated chest X-ray radiology reports. The core idea is to use a fact-checking model (FC) to identify incorrect or hallucinated findings in automatically generated reports and then correct those findings using a prompt-based LLM approach.

**Strengths:**

1. The work addresses a critical issue: factual errors in AI-generated radiology reports, which hinder real-world adoption.
2. Unlike many prior methods focused on improving the report generator itself, this method applies a non-invasive correction layer at inference time. This makes it usable with existing, deployed models—a highly pragmatic solution.
3. The evaluation is fairly comprehensive. The method wasn't just tested on one model. It was validated against 7 different SOTA report generators across  several public, multi-institutional datasets (e.g., ChestImaGenome, MS-CXR, VinDr-CXR)

**Weaknesses:**

1. The system can only correct findings already mentioned in the generated report. It cannot add new findings that the original generator omitted entirely.
2. The method does not address errors in severity (e.g., "mild" vs. "severe" pneumonia) or measurements. It is primarily focused on the presence, absence, and location of core findings
3. The entire pipeline depends on the accuracy of the initial FFL (finding extraction) and anatomical localization algorithms. An error in one of these upstream steps could cause the FC model to check the wrong thing, leading to an incorrect prompt and a flawed correction.
4. The system relies on bounding boxes for localization. While standard, bounding boxes are an approximation. Segmentation mask here can be a good option.
5. The entire system is built on extracting findings into the FFL (fine-grained finding patterns) format. While the paper cites high accuracy for this extraction , this format may be less expressive than other structured representations, like the entity-relation graphs produced by RadGraph. It's unclear if the FFL pattern can capture complex relationships between findings, which could be a source of error that this system might miss. Similar approach has been adopeted in [1] for reference using RadGraph:
[1] Semantic Consistency-Based Uncertainty Quantification for Factuality in Radiology Report Generation. NAACL 2025.
6. The 5 prompt templates used to guide the LLM are simple (e.g., "Remove 'X'"). This might be insufficient for very complex sentences where such an edit could break the grammatical flow.
7. There are mismatch in tenses. Some sentence is written is past vs some are written in present tense. Also, there are minor typos in the paper (e.g, would-> wold)

**Questions:**

1. Could the authors elaborate on the technical challenges of adding an "omission detection and insertion" module? For instance, could the FC model be used to check for a set of expected findings (beyond just those mentioned) and identify high-confidence omissions? How might the prompt generation (Table 2)  be adapted to insert a completely new finding or sentence, rather than just edit an existing one?

2. The paper notes that errors in severity (e.g., "mild" vs. "severe") and measurements are out of scope. : Is this a fundamental limitation of the chosen FFL (fine-grained finding pattern) representation, which seems to focus on the "core finding name" and "laterality"? Or is it a limitation of the FC model's training?

3. Was the choice of bounding boxes purely for convenience (e.g., availability of datasets like ChestImaGenome ), or are there architectural reasons?

4. While the examples in Table 4 are effective, complex radiology sentences can be brittle. Have the authors analyzed the failure rate of this specific LLM correction step? For instance, in what percentage of cases does the LLM, when given a simple edit prompt, produce a corrected sentence that is grammatically incorrect or semantically nonsensical, even if it successfully executes the requested edit?

---

### Official Review · Reviewer_746C · 2025-10-31

**Soundness:** 2
**Presentation:** 1
**Contribution:** 1
**Rating:** 0
**Confidence:** 5

**Summary:**

This paper introduces an inference-time method to improve the accuracy of AI-generated chest X-ray reports by detecting and correcting factual and location errors without retraining the original model. The proposed method first parses the generated report into structured fine-grained clinical findings and then builds a large synthetic dataset to model realistic findings and locations. A multimodal fact-checking (FC) network evaluates whether each reported finding is supported by the image and whether its anatomical location is consistent. When discrepancies are detected with high confidence, the system applies rule-based prompts to a lightweight language model to make minimal and focused edits to the original sentences, rather than rewriting the full report. This correction strategy reduces the risk of introducing new errors and preserves the valid content generated by the original model. Experiments across multiple datasets and different radiology report generators demonstrate consistent improvements in clinical correctness and grounding metrics.

**Strengths:**

* The proposed model works at inference time, which requires no retraining or modification of the original report-generation model.

* The multimodal fact-checking model checks whether each reported finding is present in the image and whether its stated anatomical location is correct.

* Generates a large synthetic dataset that mimics realistic reporting mistakes, enabling robust training of the correction model.

* Shows reliable performance gains across different datasets and a variety of report-generation systems.

**Weaknesses:**

* The authors claimed their method was novel which is not true. The method closely follows the architecture and pipeline introduced in the MICCAI 2025 phrase-grounded fact-checking paper (https://papers.miccai.org/miccai-2025/paper/3526_paper.pdf ). The system structure and schematic design appear highly similar, which raises questions about the level of architectural novelty. The MICCAI paper has not been cited and discussed.

* Figure 3 appears taken from the MICCAI 2025 framework, suggesting that the main contribution was already published before this submission. Therefore, this paper doesn't provide any substantial insights into the methodological direction.

* The primary difference from the MICCAI work seems to be the addition of an LLM-based correction module driven by the fact-checking output. While this is useful, it may be viewed as a minor extension of prior work rather than a fundamentally new method.

* There's no radiologist or human evaluation performed, making it difficult to judge the true clinical relevance of the corrections and how meaningful the improvements are in practice.

**Questions:**

* Can the authors clearly articulate the conceptual and architectural differences between your framework and the MICCAI 2025 phrase-grounded fact-checking system? Specifically, what new capabilities or insights are introduced beyond adding an LLM-based correction module?

* Can the authors report statistics on how often the LLM is triggered and the distribution of correction types? Which correction types helped more to improve the performance?

* Have the authors considered including radiologists' or clinicians' review to assess the clinical meaningfulness of corrections?

**Details Of Ethics Concerns:**

The authors claimed their method was novel which is not true. The method closely follows the architecture and pipeline introduced in the MICCAI 2025 phrase-grounded fact-checking paper (https://papers.miccai.org/miccai-2025/paper/3526_paper.pdf ). The system structure and schematic design appear highly similar, which raises questions about the level of architectural novelty. The MICCAI paper has not been cited and discussed. Figure 3 appears taken from the MICCAI 2025 framework, suggesting that the main contribution was already published before this submission.

---

### Official Review · Reviewer_4eQq · 2025-11-01

**Soundness:** 1
**Presentation:** 1
**Contribution:** 1
**Rating:** 0
**Confidence:** 4

**Summary:**

The paper proposes an automatic report correction pipeline which utilizes a pattern matching and a localization algorithm on findings. A trained fact-checking model verifies errors and an LLM is used to rewrite the report after the error analysis.

**Strengths:**

- The pipeline shows metric improvements which highlights the weaknesses of the current automated report generators.

**Weaknesses:**

- The writing is very difficult to follow.
- The proposed pipeline looks engineered, composing of many components that makes sense but not fully justified. It looks hard to generalize the findings.

**Questions:**

- The pipeline seems to rely on many components being good. An ablation study could show how critical each component is and whether the pipeline can be simplified.

**Details Of Ethics Concerns:**

-

---

### Official Review · Reviewer_WaKm · 2025-11-01

**Soundness:** 3
**Presentation:** 3
**Contribution:** 3
**Rating:** 4
**Confidence:** 3

**Summary:**

This paper introduces an inference-time correction framework for automatically generated chest X-ray radiology reports. The proposed method combines a fact-checking model with a LLM that performs selective sentence-level corrections based on FC-guided prompts. The approach identifies factual and spatial inconsistencies between generated reports and image evidence, and modifies only erroneous findings instead of rewriting entire sentences. Experiments show consistent quality improvements of 17–30% in report-ground-truth similarity metrics.

**Strengths:**

-inference-time correction combining vision–language verification with LLM-driven editing is pretty neat
- Well-designed experiments across multiple datasets and baselines
- Strong motivation, transparent architecture and dataset description

**Weaknesses:**

- The system handles only presence/absence and localization errors, omitting severity, measurement, and omission (missing-finding) errors
- training on synthetic error patterns may not fully capture real generator error distributions. Validation on more realistic or human-annotated error sets would strengthen claims.
- No statistical significance reported, some results (BLEU/SBERT) may have limited clinical interpretability
- I feel like there are more and better metrics that you could compare to? what about the standard clinical metrics? Things like SRRBert, F1Chexbert, GREEN, RadCliq, FineRadScore, RateScore etc. I think ReXrank and RadEval provides a good summary of things to compare to.

**Questions:**

- How sensitive is the correction accuracy to the IOU threshold and the veracity label Ep from the FC model?
- Did you evaluate inter-run variability in LLM corrections
- Can the RadCheck synthetic data be extended to modalities beyond chest X-ray?
- Why did you pick these metrics as opposed to others?

---

### Note · Authors · 2025-12-14

**Comment:**

We will bring this as a journal version in future to present the comprehensive perspective

**Withdrawal Confirmation:**

I have read and agree with the venue's withdrawal policy on behalf of myself and my co-authors.